# Fabrication, DFT Calculation, and Molecular Docking of Two Fe(III) Imine Chelates as Anti-COVID-19 and Pharmaceutical Drug Candidate

**DOI:** 10.3390/ijms23073994

**Published:** 2022-04-03

**Authors:** Hany M. Abd El-Lateef, Mai M. Khalaf, Mohamed R. Shehata, Ahmed M. Abu-Dief

**Affiliations:** 1Department of Chemistry, College of Science, King Faisal University, P.O. Box 400, Al-Ahsa 31982, Saudi Arabia; mmkali@kfu.edu.sa; 2Chemistry Department, Faculty of Science, Sohag University, Sohag 82534, Egypt; 3Chemistry Department, Faculty of Science, Cairo University, Giza P.O. Box 12613, Egypt; mrshehata_05@hotmail.com; 4Chemistry Department, College of Science, Taibah University, Madinah P.O. Box 344, Saudi Arabia

**Keywords:** tetra-dentate Schiff base, DFT calculation, antimicrobial, anticancer, antioxidant, COVID-19, molecular docking insights

## Abstract

Two tetradentate dibasic chelating Schiff base iron (III) chelates were prepared from the reaction of 2,2′-((1E,1′E)-(1,2-phenylenebis(azanylylidene))bis(methanylylidene))bis(4-bromophenol) (PDBS) and 2,2′-((1E,1′E)-((4-chloro-1,2-phenylene)bis(azanylylidene))-bis(methanylylidene))bis(4-bromophenol) (CPBS) with Fe^3+^ ions. The prepared complexes were fully characterized with spectral and physicochemical tools such as IR, NMR, CHN analysis, TGA, UV-visible spectra, and magnetic moment measurements. Moreover, geometry optimizations for the synthesized ligands and complexes were conducted using the Gaussian09 program through the DFT approach, to find the best structures and key parameters. The prepared compounds were tested as antimicrobial agents against selected strains of bacteria and fungi. The results suggests that the CPBSFe complex has the highest activity, which is close to the reference. An MTT assay was used to screen the newly synthesized compounds against a variety of cell lines, including colon cancer cells, hepatic cellular carcinoma cells, and breast carcinoma cells. The results are expressed by IC50 value, in which the 48 µg/mL value of the CPBSFe complex indicates its success as a potential anticancer agent. The antioxidant behavior of the two imine chelates was studied by DPPH assay. All the tested imine complexes show potent antioxidant activity compared to the standard Vitamin C. Furthermore, the in vitro assay and the mechanism of binding and interaction efficiency of the tested samples with the receptor of COVID-19 core protease viral protein (PDB ID: 6lu7) and the receptor of Gram-negative bacteria (Escherichia coli, PDB ID: 1fj4) were investigated using molecular docking experiments.

## 1. Introduction

Metal-based therapies have become a feasible area of research in medicinal chemistry after the unexpected discovery of cis-platin [1]. Platinum-based medications, however, are linked to (i) bad side effects, (ii) a lack of selectivity, and (iii) inherent or acquired resistance, prompting the quest for effective non-platinum therapies [2]. In coordination chemistry, tetradentate imine chelates play a significant role in materials science, biochemistry, and catalysis [3,4,5,6,7,8]. Metallosalens are bioinorganic and medicinal coordination molecules having a wide range of applications in bioinorganic and medicinal chemistry, including enzyme mimics, sensing, bioimaging, and medicine [9,10]. Imine metal chelates, in particular, are powerful physiologically active compounds with antibacterial, antifungal, antitumor, and anti-inflammatory properties [11,12,13]. Among all imines, ‘salen-type’ imines are possibly the most prevalent. The di-negative anionic forms of the salen-type imines act as tetra-dentate chelating ligands with di-oxygen and di-nitrogen atoms as donor sites to form mononuclear metal chelates with d-orbital metal ions generally. Researchers’ continued interest in the Schiff base chemistry of metal complexes has resulted in the development of novel viable therapeutic alternatives for the treatment of a variety of illnesses and malignancies. Synthetic chemists were drawn to such molecules because of their broad biological and pharmacological responses, which prompted them to design and build new scaffolds with improved features such as antibacterial, antifungal, antiviral, and anticancer qualities [14]. Spike protein interaction and inhibiting coronavirus genetic material replication are critical for a substance to be considered an anti-COVID-19 medication [15].

Many compounds are now being tested in clinical trials; nevertheless, the optimum therapy choice is still unknown. Apart from being a potential treatment for COVID-19, medication metal chelates could provide other benefits. For example, the human body requires metal ions to fulfill critical processes such as carrying oxygen to all body tissues via an iron-containing protein (hemoglobin).

Different approaches to the use of drug metal complexes as effective therapeutic agents against COVID-19 can be studied in this regard. There are numerous computational methods available that are frequently used in research to estimate the activity of chemical compounds based on knowledge of the medication and its target structure. As a result, imine complexed bioessential metals could play a key role in the development of metallodrug-based treatments for this infection.

In this regard, our group is also working on the synthesis and characterization of imine metal chelates with biomedical and industrial applications [16,17,18,19,20,21,22]. Therefore, this research compiles the synthesis of two derivatives of tetra-dentate salen ligands and their Fe^3+^ chelates. Analytically, the prepared salen ligands were identified using elemental analysis and a variety of spectrum methods, including IR, ^1^H NMR, ^13^C NMR, and UV-vis. The complexes were analyzed by other additive methods such as molar conductance, magnetic susceptibility, and thermogravimetric analysis for geometry characterization. Moreover, the stoichiometry of the prepared complexes was verified by reactions in solutions via continuous and molar ratio methods. This was followed by a geometry optimization study, to obtain important features. The tetradentate salen ligands and their Fe^3+^ chelates were evaluated in vitro for their antimicrobial, antitumor, and antioxidant activity.

## 2. Results and Discussion

### 2.1. NMR Spectra of PDBS and CPBS Imine Ligands

The ^1^HNMR data of PDBS and CPBS imine ligands (Experimental Section and Appendix A) show singlet signals at δ = 11.75 and (12.75, 12.64) that are assigned to the two phenolic –OH groups, respectively. The data also show single signals at δ = 8.69 ppm and (8.95, 8.93 ppm), which referred to the azomethine group (CH=N) proton of the PDBS and CPBS imine ligands, respectively. Furthermore, the data show singlet and doublet signals at δ = 7.90 – 6.70 ppm for aromatic protons.

In the ^13^C {1 H} NMR spectra of PDBS and CPBS imine ligands, there are signals for the CH=N at 159.96 and (163.62, 162.86 ppm). Moreover, the signals at 158.89 and 159.86 correspond to -C-OH of PDBS and CPBS imine ligands, respectively.

### 2.2. Preliminary Investigation of the Prepared PDBSFe and CPBSFe Chelates

The composition of the investigated Fe (III) complexes was resolved by elemental examination. The proportion of carbon, hydrogen, and nitrogen in complexes was established through elemental analysis. The results of this elemental analysis matched theoretically determined values quite well. Table 1 summarizes exploratory observations (color, melting point, molar conductance, and magnetic moments) as well as elemental composition.

The molar conductance of the prepared PDBSFe and CPBSFe chelates was evaluated and values were found to be 16.54 and 12.35 Ω^−1^cm^2^mol^−1^, respectively (Table 1), suggesting their non-electrolyte nature [23,24]. Thus, the nitrate group remains in the coordination sphere, as evidenced by the low values of molar conductivity [23,24].

### 2.3. Vibrational Spectral Analysis

The most prominent IR bands detected for the titled PDBS and CPBS imine ligands and Fe^3+^ over the range 4000–400 cm^−1^ with the plausible assignment are given in Table 1 and Appendix A. Comparison between vibrational absorption frequencies of PDBS and CPBS ligands and their Fe (III) imine chelates assists in comprehending the ligand’s binding mechanisms to the metal.

The bands appearing at 1612 and 1635 cm^−1^ in PDBS and CPBS ligands, respectively, are due to ν(C=N) vibration, which shifted to a lower frequency region (1602, 1608 cm^−1^) in the PDBSFe and CPBSFe chelates. This appearance justified the involvement of nitrogen in bonding with the metal centers [25,26]. Bands at 3385 and 3425 cm^−1^ were assigned to ν(O-H) vibration in PDBS and CPBS ligands, respectively, which disappeared in PDBSFe and CPBSFe chelates due to deprotonation during the complexation process. Around this region, there are peaks in the 3415, 3471 cm^−1^ region assigned to the ν(O-H) vibration of crystal water molecules with PDBSFe and CPBSFe chelates. The presence of coordinating water molecules was also inferred by the appearance of r(H_2_O) and w(H_2_O) vibrations, which were later determined by TGA analysis [27]. The phenolic (C-O) stretching vibration appears at 1273, 1279 cm^−1^ for the free PDBS and CPBS ligand, shifted towards lower frequencies (1232, 1263 cm^−1^) for PDBSFe and CPBSFe complexes, respectively. This shift demonstrates that the phenolic group’s oxygen atom is involved in the chelation with C–O–M [25,26].

The bands in the region of 590, 545 cm^−1^, 491, and 503 cm^−1^ could be attributed to ν(M ← O) and ν(M ← N) vibrations in PDBSFe and CPBSFe chelates, respectively [25,26]. The presence of a coordinated nitrate group was confirmed by three nondegenerate modes at 1451, 1477 cm^−1^ (NO_2_)_asy_, 1366, 1369 cm^−1^ (NO_2_)_sy_, and 820, 816 cm^−1^ (NO) in both PDBSFe and CPBSFe chelates, respectively [28,29].

### 2.4. Molecular Electronic Spectra and Magnetic Susceptibility Observations

UV–vis spectroscopy is the most common approach for monitoring electronic transitions inside d-block ions, especially after complexation, to ascertain the geometry obtained. Noteworthy transitions and quantitative molar absorptivity (ε_max_) are summarized in Appendix A. Concerning free PDBS and CPBS ligand spectra and their complexes (cf. Figure 1), the transitions appeared at 238–254 nm, which is attributed to π → π*, whereas at 292–398 it is attributed to n → π*. An observable shift of the n → π* band appeared more clearly in both PDBSFe and CPBSFe complexes than that in their corresponding ligands. This demonstrates changes upon lone pairs of electrons over donor atoms that were coordinated [30]. Moreover, the absorption band in the visible area was detected at 384, 440 nm, which is ascribed to the intraligand band of PDBS and CPBS compounds, respectively. Furthermore, the absorption band identified between 478 and 366 nm could be attributed to the metal-ligand charge transfer (MLCT) phenomenon for PDBSFe and CPBSFe chelates, respectively [31]. Within the 512 and 530 nm regions, strong d–d transitions in the investigated PDBSFe and CPBSFe chelates, respectively, were observed. Furthermore, the magnetic moment (μ_eff_) of the investigated complexes (Table 1) was calculated to confirm the structural formula proposed. High spin octahedral geometry is assigned to the PDBSFe and CPBSFe chelates (μ_eff_ = 5.52, 5.47 B.M., respectively) [32].

### 2.5. Thermal Analysis and Kinetics

From ambient temperature to 700 °C, a thermal study was performed to explore the likely decomposition routes in the examined PDBSFe and CPBSFe chelates. The suggested decomposition paths for the complexes are aggregated in Table 2. For instance, the decomposition behavior in the TGA curve of the PDBSFe complex was as follows: (I) within the temperature range of 35–120 °C, the first stage of degradation resulted in the removal of two hydration water molecules, leading to a 5.54 percent weight loss (calc. 5.59 percent). (II) The second phase involved removing one coordinated water molecule at 125–145 °C, which resulted in a 2.86 percent weight decrease (calc. 2.80 percent). (III) The third step within the range 150–200 °C corresponds to the removal of the coordinated nitro group 9.59% (calc. 9.63%). The fourth, fifth, and sixth stages were recorded within the range 205–650 °C, corresponding to the loss of C6H3Br, C_6_H_3_BrO, and C_8_H_6_N_2_ fragments, respectively, leaving FeO as residue [32,33]. The same behavior was noted for the CPBSFe complex.

The parameters belonging to the kinetic and thermodynamic degradation stages were calculated for the PDBSFe and CPBSFe chelates. The results are shown in Table 2, which reveals the following observations.

1—In all cases, first-order decomposition was the best fit for all decomposition stages of complexes. 2—Negative activation entropy ΔS values could be ascribed to the activated complex, which is more than reactants [22,34]. 3—The endothermic nature of decomposition processes is denoted by the positive value of ΔH [35]. 4—The decrease in the removal rate from one decomposition stage to the next is shown by the increase in the value of ∆G. 5—Furthermore, the positive sign of G indicates that the number of final residues was greater than in the early stage, indicating that the decomposition step was not spontaneous.

### 2.6. Stoichiometry, Formation Constants, and pH Range of Stability

Stoichiometry of the investigated PDBSFe and CPBSFe chelates that formed in solution between PDBS or CPBS ligands and metal salts (Fe(III)) was evaluated by continuous variations and molar ratio approaches. The continuous variation and molar approaches yielded curves (Figure 2 and Appendix A), showing maximum absorbance at a mole fraction equal to 1Fe^3+^:1L in the molar ratio for PDBS or CPBS ligands. Although there is an acceptable difference in complex composition whether created in solid-state or in solution, consistency improves in the solid-state composition. Using the continuous variation approach, the formation constant (K_f_) of each complex was also computed (Table 3). As a consequence, the following is presented as the order of stability of the complexes; PDBSFe > CPBSFe complex. Gibb’s free energy values were also obtained to demonstrate that the metal chelate formation is spontaneous [23,24,33,35]. According to all practical techniques, the most probable complex formulae are schematically displayed in Figure 1.

The pH stability profiles estimated for the complexes exhibited an obvious similarity between the complexes (Figure 3) that were fairly stable along with the pH = 4.5–11 range. The fact that there is a noticeably large range of pHs reflects their stability under various pHs, so different applications can be performed safely without infection [36,37].

### 2.7. Molecular Optimization of the Prepared PDBS and CPBS Imine Ligands and Their Fe^3+^ Chelates

#### 2.7.1. Molecular DFT Calculation of PDBS and CPBS Imine Ligands

Figure 4 depicts the ligands’ optimized structures as the lowest energy configurations. Natural bond orbital analysis (NBO) shows that the active sites have higher negative charges than the inactive sites: O_1_ (−0.675), O_2_ (−0.675), N_1_ (−0.536), and N_2_ (−0.538) for PDBS; and O_1_ (−0.669), O_2_ (−0.669), N_1_ (−0.546), and N_2_ (−0.544) for CPBS. The Fe^3+^ ions form tetra-dentate coordination to O_1_, O_3_, N_1_, and N_2_ with one 5- and two 6-member rings.

#### 2.7.2. Molecular DFT Calculation of [FePDBS(H_2_O)NO_3_] and [FeCPBS(H_2_O)NO_3_]

Figure 5 shows the optimized structures of the complexes [FePDBS (H_2_O) NO_3_] and [FeCPBS (H_2_O) NO_3_] as the lowest energy configurations. The iron atom is six-coordinate in an octahedral geometry with ONO2 and O3 of the coordinated water molecule in axial position and atoms N1, N2, O2, and O_1_ are almost in one plane deviated by 2.628° and 1.719° for [FePDBS(H_2_O)NO_3_] and [FeCPBS(H_2_O)NO_3_], respectively. The bite angles N1-Fe-N2 are 84.35 and 84.54 and are lower than 90° due to chelation for [FePDBS(H_2_O)NO_3_] and [FeCPBS(H_2_O)NO_3_], respectively. The bond angles in the square range from 84.35° to 92.39°; see Table 4.

On the coordinated atoms, the natural charges estimated from the NBO analysis are Fe (+0.667), N1 (−0.407), N2 (−0.411), O1 (−0.597), O2 (−0.626), O3 (−0.794), and O4 (−0.415) for [FePDBS(H_2_O)NO_3_]; and Fe (+0.664), N1 (−0.414), N_2_ (−0.412), O1 (−0.591), O2 (−0.622), O_3_ (−0.795), and O_4_ (−0.412) for [FeCPBS(H_2_O)NO_3_]; see Figure 5.

#### 2.7.3. Physical Characteristics for the Prepared Compounds

Table 5 shows the predicted total energy, highest occupied molecular orbital (HOMO) energies, lowest unoccupied molecular orbital (LUMO) energies, and dipole moment for the titled imine ligands and their Fe(III) chelates. The total energy of Fe(III) chelates is more negative than that of free ligands, indicating that complexes are more stable than free ligands. The most significant orbitals in a molecule are the frontier molecular orbitals, HOMO and LUMO. The HOMO energy describes the electron-donating ability, while LUMO energy describes the ability of electron acceptance for the compound. The HOMO helps to describe the chemical reactivity and kinetic stability of the molecule. The energy gaps (Eg) = E_LUMO_ − E_HOMO_ are smaller in the case of complexes than that of the ligand due to ligand-to-metal-ion chelation; see Table 5 and Figure 6. The lowering of Eg in complexes compared to that of ligands explains the charge transfer interactions upon complex formation. The polarity of the complex is much larger than the free ligand.

The ionization energy, I, electron affinity, A, electronegativity, χ, global softness, S, chemical hardness, η, chemical potential, μ, and electrophilicity were estimated for the ligands and complexes.

### 2.8. Pharmacological Studies

#### 2.8.1. Antimicrobial Activity

The antimicrobial effects of the prepared bis-hydrazone ligand and its metal complexes were investigated for selected strains of bacteria (*S. marcescence*, *M. luteus*, *E. coli*) and fungi (*A. flavus*, *C. candidum*, *F. oxysporum*) (Appendix A and Figure 7), while Ofloxacin and Fluconazole were used as guidelines for antibacterial and anti-fungal behavior, respectively. The CBPSFe complex displayed the highest inhibitory effect on all microorganisms under investigation based on the estimated inhibition diameter (mm/µg Sample). There were also enhancements in the antimicrobial efficiency for the complexes compared to its PDBS and CPBS imine ligands due to the chelation theory [36,37,38,39]. The partial sharing of its positive charge with the hetero donor atoms of the ligand, as well as electron delocalization over the entire chelate ring system, significantly diminish the polarity of the metal ion during chelation [40,41]. Various antibacterial potency can be due to variations in the composition of the cell wall of the microorganisms [42,43]. Inhibition zone data for the prepared compounds were confirmed by calculating the activity index as seen in Appendix A according to the following equation [44,45].
(1)$$\;\mathrm{Activity}\;\mathrm{index}(A)=\frac{\mathrm{inhibition}\;\mathrm{zone}\;\mathrm{of}\;\mathrm{complex}(\mathrm{mm})}{\mathrm{inhibition}\;\mathrm{zone}\;\mathrm{of}\;\mathrm{standard}\;\mathrm{drug}(\mathrm{mm})}\times 100$$

##### Determination of Minimum Inhibition Concentration

The findings of the serial dilution (MIC) correspond well with those of the will diffusion experiment, as shown in Table 6. In comparison to other complexes, the CBPSFe complex (MIC: 3.25, 3.50, and 2.50 µg/mL) has shown superior bioactivity against *S. marcescence*, *E. coli*, and *M. luteus*. The same behavior was noted for the tested fungi. *M. luteus* was found to be the most sensitive bacteria among all bacteria involved in the study, whereas *E. coli* was the most resistant. Particularly, *Getrichm candidum* appeared as the most sensitive fungus among all fungi involved in this study.

#### 2.8.2. Anti-Cancer Activity

The in vitro cytotoxic effect of the free BS ligand and its synthesized complexes was assessed against targeted cancer cell lines at different concentrations (0, 0.1, 0.5, 1, 2, 5, 10, and 50 μg/mL) for 24 and 48 h. The dosage of DMSO used to dissolve the compounds was below the cytotoxicity threshold (less than the 0.1 percent range as suggested for the cell culture). According to the dose applied, the anticancer activity of investigated compounds increased by increasing their concentrations. These tested compounds show an impact on breast carcinoma cells. CPBSFe complex grabs the superiority by being active at a concentration of 5.74 µg/mL (Table 7). From IC50 values, we can apply the CPBSFe complex as a drug candidate for tumors. The cytotoxicity of the prepared compounds’ results in Table 7 indicates the mean numbers ± SD of three independent biological experiments. Metal chelates’ cytotoxicity is assumed to be based on their capacity to bind DNA and so disrupt its structure, causing replication and transcription processes to be inhibited, and finally cell death [46,47].

#### 2.8.3. Antioxidant Activities

The antioxidant activity of the investigated compounds in the DMSO medium was determined by the DPPH method. The color of the free DPPH radical is deep purple, and with the introduction of each compound, the color changes to yellow [48]. This finding indicates that the PDBS and CPBS imine ligands and their Fe^3+^ chelates have higher antioxidant action. The percentages of inhibitions for the prepared compounds are shown in Figure 8. This conclusion was further corroborated by the fact that the produced complexes have stronger antioxidant activities against the DPPH free radical than the standard Vitamin C.

### 2.9. Molecular Docking Insights of the Compounds under Investigation against Escherichia coli Bacteria

The free energy of binding for the PDBS, CPBS imine ligands, and their Fe (III) chelates with the receptor of Escherichia coli protein (PDB ID: 1fj4) were found to be −4.9, −6.6, −8.3, and −10.3 kcal/mol for PDBS and CPBS, [FePDBS(H_2_O) NO_3_] and [FeCPBS(H_2_O)NO_3_], respectively; see Table 8. The more negative the binding energy, the greater the association. Therefore, the interactions are in the order of PDBS ˂ CPBS ˂ [FePDBS(H_2_O)NO_3_] ˂ [FeCPBS(H_2_O)NO_3_].

The 2D and 3D plots of the interaction of PDBS, CPBS, [FePDBS(H_2_O)NO_3_], and [FeCPBS(H_2_O)NO_3_] with the active site of the Escherichia coli protein receptor (PDB ID are shown in Figure 9, Figure 10, Figure 11 and Figure 12.

### 2.10. Docking Studies of the Prepared Compounds against COVID-19

The binding free energies of ligand and complex with the COVID-19 receptor main protease viral protein (PDB ID: 6lu7) were found to be −4.6, −6.9, −13.3, and −18.0 kcal/mol for PDBS, CPBS, [FePDBS (H_2_O) NO_3_], and [FeCPBS (H_2_O) NO_3_], respectively; see Table 9. The more negative the binding energy, the greater the association. Therefore, the interactions are in the sequence of PDBS ˂ CPBS ˂ [FePDBS(H_2_O)NO_3_] ˂ [FeCPBS(H_2_O)NO_3_].

The 2D and 3D plots of the interaction of PDBS, CPBS, [FePDBS(H_2_O) NO_3_], and [FeCPBS(H_2_O)NO_3_] with the active site of the COVID-19 receptor main protease viral protein (PDB ID: 6lu7) are shown in Figure 1 and Figure 2.

## 3. Materials and Methods

### 3.1. Materials

Starting reagents were supplied by Aldrich and all manipulations were carried out with the materials as acquired.

### 3.2. Instruments

The melting points of the investigated ligands and decomposition points of their Fe(III) chelates were taken on a Mel-Temp, using a capillary melting point. Infrared spectra (ν/cm^−1^) were obtained using an FT-IR Agilent Technology spectrophotometer model Cary 630 (at 298 K) in the range 4000–400 cm^−1^. ^1^H-NMR was recorded using a 400 MHz NMR Spectrometer in DMSO. Carbon, hydrogen, and nitrogen microanalyses were performed at Cairo University’s Microanalytical Center utilizing a CHNS-932 (LECO) Vario Elemental Analyzer. The absorbance was measured on a UV-vis spectrophotometer. Other equipment used comprised rotary evaporator apparatus, using a retort stand, reflux system, weighing balance, filtration system with suction, and glassware. The molar conductance of the prepared complexes (1.0 × 10^−3^ mol dm^−3^) was determined in EtOH at a temperature of 25 °C using a Jenway conductivity meter instrument. The magnetic moment was evaluated via Gouy magnetic susceptibility balance at room temperature using CuSO_4_ with 5H_2_O as a calibrator. TGA, thermogravimetric analysis, was demonstrated for the investigated iron (III) chelates in the presence of nitrogen using a TGA-50 H Shimadzu thermal analyzer.

### 3.3. Preparation of PDBS and CPBS Imine Ligands

PDBS and CPBS imine ligands were synthesized according to the following procedures: 4.04 g of 5-bromo-2-hydroxybenzaldehyde (20 mmol) was dissolved in 40 mL of EtOH and mixed with 1,2-phenylenediamine (1.08 g, 10 mmol) or 4-choloro-l,2-phenylenediamine (1.43 g, 10 mmol), which was dissolved in 30 mL of EtOH. The mixture of both solutions was refluxed for 4 h, and during that time, yellow and orange precipitates formed.

PDBS ligand: ^1^H NMR (ppm): δ = 11.75 (S, 2H, -OH); δ =8.69 (S, 2H, -N = CH-); δ = 7.62 (S, 2H, -CH aromatic of ortho position with C-Br); 7.33 (d, 2H, -CH- ortho position with C-Br); δ = 7.34, 7.28 (t, 4H, -CH aromatic); 6.70 (d, 2H, -CH- ortho position with C-OH).

C^13^ NMR (ppm): δ = 159.96 (2-C = N) 158.89 (2C-OH), 139.96; 136.43; 133.66; 123.95; 120.38; 120.09; 112.20.

CPBS ligand: ^1^H NMR (ppm): δ = 12.75 (S, H, -OH); 12.64 (S, H, -OH); δ = 8.95, 8.93 (S, 2H, -N = CH-); δ = 7.90-7.89 (d, 2H, -CH aromatic adjacent C-Br); δ = 7.60 (S, 2H, -CH aromatic of ortho position with C-Br); δ = 7.60-7.47 (d, 2H, -CH aromatic ortho position of C-Cl); δ = 6.97-6.95 (d, 1H, CH aromatic of meta position of C-Cl); 7.65-7.53 (d,2H,-CH- ortho position with C-Br); 6.97-6.95 (d, 2H, -CH- ortho position with C-OH).

C^13^ NMR (ppm): δ = 163.62; 162.86 (2-C = N) 159.86 (2C-OH), 143.86; 141.51; 136.47; 136.31; 134.20; 134.11; 132.47; 128.06; 121.96; 121.91; 121.65; 120.09; 119.69; 119.67; 110.50.

### 3.4. Synthesis of PDBSFe and CPBSFe Imine Chelates

PDBS Fe and CPBSFe imine chelates were prepared according to the following procedures: (1.90 g 4 mmol) of PDBS ligand or (2.03 g 4 mmol) of CPBS ligand in 30 mL EtOH was added to 1.62 g Fe(NO_3_)_3_. 9H_2_O (4 mmol) dissolved in 15 mL of water was added slowly to the initial mixture. The resulting solution was stirred under reflux for 2 hours. The resulting brown precipitates were filtered, washed with H_2_O, and dried with Et_2_O. Yield: PDBS Fe, 91%; CPBSFe, 86%.

### 3.5. Kinetic Studies of Prepared MABS Fe and NABS Fe Imine Complexes

The Coats–Redfern relationship was used to compute the kinetic parameters of the complexes’ degrading process using the equation below [49,50,51]:(2)ln[ln(w∞/(w∞−w))T2]=ln[AR∅E∗(1−2RTE≠)]−E≠R1T

The enthalpy (∆*H*^≠^), the entropy (∆*S*^≠^), and the free energy change of activation (*G*^≠^) were computed from the equations below [23,24,25]:(3)$$\Delta H^{\neq }=E^{\neq }-RT$$
(4)ΔS≠=2.303R log (AhkbT)
(5)$$\Delta G^{\neq }=\Delta H^{\neq }-T\Delta S$$

### 3.6. Complexation Nature in Solution

Fe^3+^ chelates were evaluated in solution for stoichiometry and stability. This can be accomplished using Job’s (continuous variation) method and the mole ratio approach [52,53,54]. In equilibrated mixes of PDBS and CPBS imine ligands and Fe^3+^ ions, the optical density was measured. Subsequently, for each solution, the resulting absorbance was charted against the mole fraction of metal ([L/[L] + [M]) or the molar ratio ([L]/[M]).

The formation constant for the prepared imine Fe(III) chelates was calculated according to the following equation [52,53,54]
(6)Kf=AAm(1−AAm)2C

The free energy change, Δ(G*), of the complexes was determined at 25 °C according the following equation
Δ(G*) = −RT ln K_f_(7)
where K_f_ is the formation constant, R is the gas constant, and T is the temperature in kelvin.

### 3.7. DFT Calculation

Using the Gaussian 09 program, density functional theory (DFT) simulations were performed to study the equilibrium structure of PDBS and CPBS imine ligands and their Fe^3+^ chelates [55] at B3LYP/6-311G++(dp) for all atoms except Fe at the B3LYP/LANL2DZ level of theory for the ligands and their Fe^3+^ chelates.

### 3.8. Bioactivity

#### 3.8.1. Anti-Pathogenic Activity

The Well diffusion approach [32,56,57] was used to test in vitro antifungal and antibacterial activities of PDBS and CPBS imine ligands and their Fe^3+^ chelates. DMSO was used to prepare stock solutions of different concentrations (µg/mL) to evaluate antimicrobial efficiency. The medium for antifungal activity was prepared by adding 65 g of Sabround Dextrose Agar (SDA) to 1 L of distilled water (DW). The mixture of SDA and DW was heated with stirring to form a uniform solution. The sterilization was performed by autoclaving this solution (media) for 15 min at 121 °C. The media were spread uniformly on sterile Petri plates having a diameter of 90 mm and kept at rest until solidification. The same procedure was followed to prepare nutrient agar (NA) medium i.e., 28 g NA in 1 L of DW to explore antibacterial activity. After solidification, four wells (4 mm diameter) were taken out through sterile cork from the solidified media. The suspension form of the tested fungi was spread over the surface and the separate wells were filled with different concentration stock solutions. Finally, the Petri plates were sealed and stored at low temperature for 2 to 3 h for diffusion and then incubated at room temperature (27 °C) for 24 h, after which the inhibition zones were measured (mm).

#### 3.8.2. Anticancer Activity

To assess the cytotoxicity of the prepared compounds, the HCT-116, HepG-2, and MCF-7 cell lines were incubated in a DMEM medium containing FBS (10 percent *v/v*), streptomycin (100 g/mL), and penicillin (100 g/mL), and maintained at 37 °C in a 5 percent CO_2_ incubator [58,59]. Following the incubation period, serial dilutions of the compounds were added to the wells in triplicate, and the incubation period was extended for another 48 h. Before 1 h from the end of the incubation, 20 µL of MTT (5 g/mL in PBS) was applied to each well. The plates were shaken after the incubation time and the supernatant liquid was extracted. Each well received 100 µL of DMSO. The optical density (OD) was measured at 540 nm after 15 min of room temperature incubation. The below formula was used to calculate the percentage of cells that were dead [60]:% Inhibition = {(Abs_control_ − Abs_sample_)/(Abs_sample_)} × 100(8)

The IC_50_ values of the compounds were generated from the dose–response curves and reported as the average of three independent experiments. The IC_50_ value was determined as the average standard deviation.

#### 3.8.3. Antioxidant Activity

Serial dilutions of PDBS and CPBS imine ligands and their Fe^3+^ chelates (25, 50,100, 200, and 400 µM) were combined with an ethanolic solution containing 85 M DPPH radical. Each mixture was kept for 30 min at room temperature and the absorbance decrease was monitored at 518 nm using a UV-vis spectrophotometer [28,29]. A positive control applied was ascorbic acid, which was used in the same concentrations as the drugs. The following formula was used to estimate the percentage inhibition of the compounds and ascorbic acid:DDPH inhibition effect (%) = (A_o_ − A_s_)/A_o_ × 100(9)

MOA2019 was used to conduct molecular docking studies [61]. The mechanism of binding and interaction efficiency of the tested samples with the receptor of COVID-19 core protease viral protein (PDB ID: 6lu7) [62] and the receptor of Gram-negative bacteria (Escherichia coli, PDB ID: 1fj4) [63] were estimated. The optimized structures of PDBS and CPBS imine ligands and their c Fe^3+^ chelates were produced in PDB file format using the outcome of Gaussian09 computations. The receptor crystal structures were retrieved from the Protein Data Bank (http://www.rcsb.org/pdb, accessed on 1 March 2022).

## 4. Conclusions

In conclusion, two salphen derivatives were designed and their Fe^3+^ chelates were prepared and characterized via various physicochemical and spectroscopic tools such as NMR, molecular electronic and vibrational spectra, CHN, TGA, and magnetic moment susceptibility. The optimizations of the structures were investigated by DFT calculations. Correlation between all experimental data shows that PDBS and CPBS imine ligands act as tetra-dentate ligands via deprotonated two-hydroxyl groups and two-azomethine groups to afford octahedral complexes with Fe^3+^ ions. Microanalyses confirmed that the complexes were developed in a 1:1 (ligand: metal) ratio. The complexes’ molar conductivity measurements in DMF demonstrated that they are non-electrolytes. Moreover, the PDBS and CPBS imine ligands and their Fe^3+^ chelates were evaluated regarding antimicrobial, anticancer, and antioxidant activity and the CBPSFe complex showed superiority over all the prepared compounds against the selected strains of bacteria, fungi, and cancer cells. Furthermore, the possible modes of binding to the most active sites of COVID-19 main protease viral protein (PDB ID: 6LU7) and of the Escherichia coli receptor (Gram-negative bacteria) (PDB ID: 1jij) were evaluated by docking studies.

## Data Availability

The raw/processed data generated in this work are available upon request from the corresponding author.

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
