# Peer review of "Fabrication, DFT Calculation, and Molecular Docking of Two Fe(III) Imine Chelates as Anti-COVID-19 and Pharmaceutical Drug Candidate"

_ijms, 2022, doi:10.3390/ijms23073994_

Round 1

Reviewer 1 Report

The author has reported SARS-CoV-2 main protease inhibitors using DFT calculation. In my opinion, the paper needs major revision.

Comments:

  1. The title is too long needs to revise. Abstract and Introduction are quite premature and shallow. Ideally, readers expect to have a very brief account of the aims, methods, key findings, and conclusions of a study from an abstract with a couple of sentences from each part.
  2. English is weak. The authors need to improve their writing style. The whole manuscript needs to be checked by native English speakers.
  3. Every section of the manuscript must be written scientifically according to the published literature with appropriate references.
  4. The logical flow of this manuscript is not perfect. The authors have written several matters haphazardly. The work appears as groundwork.
  5. Spacing, punctuation marks, grammar, and spelling errors should be reviewed wholly.
  6. The flow of the introduction is not complete and unspecific. My recommendation is to construct the sentences more lucid and legible for more productive comprehension.
  7. The first paragraph of the introduction section contains no new information. Need to change.
  8. The introduction section seems missing important information. This section needs profound modification with different medical hypotheses recommended different treatment approaches and gave convincing arguments for new discoveries urgently needed for treatment option COVID-19. Authors can also write some about the recent mutation.
  9. The resolution of the figures must be improved.
  10. The vital issue is the outlook of the figures. Need to change the docking figures to be more attractive.
  11. Page no 12, line no 328, the author needs to describe in detail the effect of LUMO and MOMO energy.

Author Response

Dear Sir: Editor of International journal of molecular sciences

First, thank you for your kind concern and cooperation for the opportunity to publish our manuscript in your respectable and high impacted Journal.

We are very excited to have been given the opportunity to revise our manuscript. We carefully considered your comments as well as those offered by the three reviewers. We want to extend our appreciation for taking the time and effort necessary to provide such insightful guidance. The revision, based on the review team’s collective input, includes a number of positive changes. Based on your guidance, we have accordingly modified the manuscript and detailed corrections, changes and/or rebuttals against each point raised are listed below.

We hope that these revisions improve the paper such that you and the reviewers now deem it worthy of publication in International journal of molecular sciences. Herein, we explain how we revised the paper based on those comments and recommendations and we offer detailed responses to your comments as well as those of the reviewers. Next, we offer detailed responses of the reviewers’ comments.

Thank you again

Ahmed M. Abu-Dief

Please, find out below the answers of all reviewers’ comments.

Reviewer #1:

  1. 1. The title is too long needs to revise. Abstract and Introduction are quite premature and shallow. Ideally, readers expect to have a very brief account of the aims, methods, key findings, and conclusions of a study from an abstract with a couple of sentences from each part.

Response: Thank you very much for your valuable comment

We made revision for all mentioned items and made the necessary changes

2-         English is weak. The authors need to improve their writing style. The whole manuscript needs to be checked by native English speakers.

Response: Thank you very much for your valuable suggestion

Thank you very much for giving us the opportunity to revise our manuscript. We have made corrections to the grammar and English usage with the help of my native English speaking teacher, which this revision can make our paper more acceptable. Moreover, based on the reviewer’s suggestion; we have carefully revised the manuscript by use of “premium Grammarly (https://app.grammarly.com/ddocs/427154829)”. The revised details can be found in the revised version.

3-         Every section of the manuscript must be written scientifically according to the published literature with appropriate references.

Response: Thank you very much for your valuable comment

We make deep discussion for each section in the manuscript according to the published literature with appropriate references.

  • The logical flow of this manuscript is not perfect. The authors have written several matters haphazardly. The work appears as groundwork.

Response: Thank you very much for your valuable comment

We did all of our best to make the manuscript is more fluent

  • Spacing, punctuation marks, grammar, and spelling errors should be reviewed wholly.

Response: Thank you very much for your valuable comment

We did all of our best to fix the mentioned problems within the manuscript. Each change was highlighted in yellow color.

  • The flow of the introduction is not complete and unspecific. My recommendation is to construct the sentences more lucid and legible for more productive comprehension.

Response: Thank you very much for your valuable comment

We constructed the sentences to be more lucid and legible for more productive comprehension.

7-The first paragraph of the introduction section contains no new information. Need to change.

Response: Thank you very much for your valuable suggestion

Ok Sir, We remove the fires paragraph and replace it by another appropriate paragraph. 

8-The introduction section seems missing important information. This section needs profound modification with different medical hypotheses recommended different treatment approaches and gave convincing arguments for new discoveries urgently needed for treatment option COVID-19. Authors can also write some about the recent mutation.

Response: Thank you very much for your valuable suggestion

We modified the introduction following your valuable instructions

  • The resolution of the figures must be improved.

Response: Thank you very much for your valuable comment

 We enhanced the resolution of all figures and prepared them as tiff format

  • The vital issue is the outlook of the figures. Need to change the docking figures to be more attractive.

Response: Thank you very much for your valuable comment

We did all of our best to make docking figures more attractive.

  • Page no 12, line no 328, the author needs to describe in detail the effect of LUMO and MOMO energy.

Response: Thank you very much for your valuable comment

We made deep discussion for the effect of LUMO and MOMO energy. The most significant orbitals in a molecule are the frontier molecular orbitals, HOMO and LUMO. The HOMO energy describes the electron donating ability, while LUMO energy describes the ability of electron acceptance for the compound. The HOMO helps to describe the chemical reactivity and kinetic stability of the molecule.

Reviewer 2 Report

This paper describes on synthesis of salen type iron complex and its biological activity.

Essentially, it should be accepted as it is, because there were no serious issue throughout the manuscript.

Before acceptance, however, please improve assumption of ligand-protein docking discussion. Namely, why axial ligands (especially ions) of the iron complex were kept after docking to protein molecule? If they left from iron, iron will have coordination sites for protein residues potentially.

This problems are said to be difficult in the field of prediction of ligand-protein docking computation or AI based drug design. Please mention authors' opinion or ways of thinking during this study. 

That's all.

Author Response

Dear Sir: Editor of International journal of molecular sciences

First, thank you for your kind concern and cooperation for the opportunity to publish our manuscript in your respectable and high impacted Journal.

We are very excited to have been given the opportunity to revise our manuscript. We carefully considered your comments as well as those offered by the three reviewers. We want to extend our appreciation for taking the time and effort necessary to provide such insightful guidance. The revision, based on the review team’s collective input, includes a number of positive changes. Based on your guidance, we have accordingly modified the manuscript and detailed corrections, changes and/or rebuttals against each point raised are listed below.

We hope that these revisions improve the paper such that you and the reviewers now deem it worthy of publication in International journal of molecular sciences. Herein, we explain how we revised the paper based on those comments and recommendations and we offer detailed responses to your comments as well as those of the reviewers. Next, we offer detailed responses of the reviewers’ comments.

Thank you again

Ahmed M. Abu-Dief

Please, find out below the answers of all reviewers’ comments.

Reviewer #2:

This paper describes on synthesis of salen type iron complex and its biological activity.

Essentially, it should be accepted as it is, because there were no serious issues throughout the manuscript.

Before acceptance, however, please improve assumption of ligand-protein docking discussion. Namely, why axial ligands (especially ions) of the iron complex were kept after docking to protein molecule? If they left from iron, iron will have coordination sites for protein residues potentially.

These problems are said to be difficult in the field of prediction of ligand-protein docking computation or AI based drug design. Please mention authors' opinion or ways of thinking during this study. 

Response: Thank you very much for your valuable positive comments on our manuscript

In docking studies, we compared the interactions between free ligands and their complexes with protein. Of course, if we remove axial molecules the interactions will increase due to the availability of more sites. If we are docking the free metal ions, we expect even more sites of coordination to the protein. But we are docking the complexes as a whole.  

Round 2

Reviewer 1 Report

The revised version of the manuscript includes all remarks and modifications indicated. The main concerns of the manuscript have been solved. In my opinion, the provided version is now suitable for publication